# Assessing Global Efforts in the Selection of Vertebrates as Umbrella Species for Conservation

**DOI:** 10.3390/biology12040509

**Published:** 2023-03-28

**Authors:** Nan Yang, Megan Price, Yu Xu, Yun Zhu, Xue Zhong, Yuehong Cheng, Bin Wang

**Affiliations:** 1Institute of Qinghai-Tibetan Plateau, Southwest Minzu University, Chengdu 610225, China; yangnan0204@126.com; 2Key Laboratory of Bio-Resource and Eco-Environment of Ministry of Education, College of Life Sciences, Sichuan University, Chengdu 610065, China; meganprice@scu.edu.cn; 3Key Laboratory of National Forestry and Grassland Administration on Biodiversity Conservation in Karst Mountainous Areas of Southwestern China, School of Life Sciences, Guizhou Normal University, Guiyang 550001, China; xuyu608@gznu.edu.cn (Y.X.); shengwu_zhuyun@126.com (Y.Z.); 4Key Laboratory of Southwest China Wildlife Resources Conservation (Ministry of Education), China West Normal University, Nanchong 637009, China; wangbin513@cwnu.edu.cn; 5Wolong National Nature Reserve Administration Bureau, Wenchuan 623006, China; cyh8155@163.com

**Keywords:** biological feature, conservation status, study effort, surrogate species, terrestrial vertebrate, umbrella-species strategy

## Abstract

**Simple Summary:**

Conservation funds and resources have long been globally inadequate, and methods that could maximize conservation outcomes with limited investment, such as umbrella-species strategies, are thus needed to address the current biodiversity crisis. In this study, we summarized 242 published scientific articles and found 213 terrestrial vertebrates that were recommended as umbrella species. We summarized global trends in umbrella species selection and research during the past four decades, with North America, Europe, and Asia over-representing umbrella-related studies, and thus, more umbrella species recommendations have occurred in the Northern Hemisphere. Generally, there has been a bias toward recommending bird and mammal species, wide-ranging species, and non-threatened species, such as umbrellas, and grouses (order Galliformes) and large carnivores have often been recommended as umbrellas across different continents by multiple studies. Given observed biases and trends, we raise concerns about neglecting amphibians and reptiles, the over-preference for wide-ranging and non-threatened species, and recommend little-known species. We argue that conservation umbrella strategies can be cost-effective and successful given that appropriate species are chosen in the right location, and our findings could inform future conservation research and practices using conservation umbrella strategies.

**Abstract:**

The umbrella-species strategy has been proposed as an attainable tool to achieve multi-species and community conservation with limited investment. There have been many umbrella-related studies since the concept’s inception; thus, a summary of global study efforts and recommended umbrella species is important for understanding advances in the field and facilitating conservation applications. Here, we collated 213 recommended umbrella species of terrestrial vertebrates from 242 scientific articles published during 1984–2021 and analyzed their geographic patterns, biological features, and conservation statuses to identify global trends in the selection of umbrella species. We found a considerable geographic bias: most studies and, consequently, recommended umbrella species are from the Northern Hemisphere. There is also a strong taxonomic bias, with grouses (order Galliformes) and large carnivores being the most popular umbrella species and amphibians and reptiles being largely overlooked. In addition, wide-ranging and non-threatened species were frequently recommended as umbrella species. Given the observed biases and trends, we caution that appropriate species need to be chosen for each location, and it is important to confirm that popular, wide-ranging species are effective umbrella species. Moreover, amphibians and reptiles should be investigated for their potential as umbrella species. The umbrella-species strategy has many strengths and, if applied appropriately, may be one of the best options in today’s conservation research and funding landscape.

## 1. Introduction

The rapid loss of global biodiversity is one of the most urgent challenges facing humanity [1]. Unfortunately, given that global conservation has long been inadequately funded, limiting outcomes [2], addressing the global biodiversity crisis is quite challenging. Therefore, it may be prudent to revisit strategies that could protect more species more economically and effectively [3]. The umbrella-species strategy has been proposed as a shortcut to achieving broader conservation aims while overcoming funding and information constraints [4,5]. Essentially, the concept is that conserving one umbrella species confers protection to a large number of co-occurring species [6,7]. Compared with the other conservation surrogates, such as flagship species (species that can be used as a symbol of conservation campaigns to attract public awareness and investment) and keystone species (species that have great impacts on many other species, communities, or ecosystems), the umbrella-species concept reflects the notions and practices of protection more directly [5,8]. Certainly, these terms often overlap in several species. For example, the charismatic giant panda (*Ailuropoda melanoleuca*) has become a successful flagship species raising public appeal, driving government policy, and attracting conservation funds throughout the world [9]; meanwhile, it has also been recommended as an umbrella species because by conserving vast areas of the intact bamboo forest it meets the giant panda’s dietary requirements and many other bamboo-dependent forest species are protected [10,11]. The umbrella effect of protecting sympatric species has been documented by numerous empirical studies, suggesting that it is an attainable and efficient strategy for maximizing conservation outcomes under limited resources [10,12,13,14].

Although there is debate as to when the umbrella-species concept was first proposed, we consider the first use of the term ‘umbrella species’ by Wilcox [15] as the key starting point. Since then, diverse criteria for the selection of umbrella species have been developed, such as selection based on body size [16], home range [17], geographic range [18], relationships with sympatric species [19], or representativeness for taxonomic and functional diversity [20]. Consequently, diverse taxa have been recommended as umbrellas, and umbrella species lists have been compiled to prioritize conservation management by governments and non-governmental organizations [21,22]. A comprehensive understanding of global research and the umbrella species they have recommended, especially their biological features and conservation statuses, is fundamental for understanding advances in this field and facilitating future conservation applications. However, no study has systematically reviewed existing umbrella species in the past nearly four decades since the concept’s inception, except for Roberge and Angelstam [6], who summarized the key studies prior to 2004.

Terrestrial biodiversity and ecosystems are highly related to sustainable development and human wellbeing. However, terrestrial vertebrates have been threatened by multiple intrinsic and extrinsic pressures and are thus at unprecedented extinction risks globally, raising concerns and demands to maximize the effectiveness and accuracy of global conservation efforts [1,23]. In this study, we scanned scientific articles focused on umbrella species published since 1984 and collated those terrestrial vertebrates (amphibians, reptiles, birds, and mammals) that were recommended as umbrella species. We aimed to determine any overall trends in the study efforts and umbrella species recommendations from a global-scale perspective by analyzing geographic patterns, biological features, and the conservation statuses of recommended umbrella species. Given the taxonomic and/or geographic imbalances in the identities of the species that were proposed as umbrella taxa could potentially limit the usefulness of this strategy and thereby undermine effective conservation policy, we then provided guidance for future conservation research and practices using this strategy based on the potential biases or inadequacies we revealed in this study.

## 2. Materials and Methods

We conducted a literature search using Web of Science^TM^ (Science Citation Index Expanded; https://www.webofscience.com/ (accessed on 1 June 2022)), with the search topic “umbrella species” and a timespan from 1984 to 2021. We included those early view articles (without an assigned volume, issue, and pages) that were published online in advance of formal publication as well. In total, we obtained 735 articles and scanned them manually. We only considered articles that explicitly recommended one or several terrestrial vertebrate species (classes Amphibia, Reptilia, Aves, and Mammalia) that were studied as umbrella species and so excluded (1) reviews and meta-analyses, (2) studies on plants and invertebrates, (3) studies on aquatic-obligative vertebrates (such as fishes and whales), and (4) studies that considered all species within a guild or taxon as an umbrella species (e.g., recommending all parrots as umbrella species). We strictly distinguished umbrella species from other conservation surrogate terms and excluded the articles that only stated their study species as flagship species, keystone species, indicator species, or other surrogates while including the articles that recommended the species as umbrella species and other surrogates simultaneously. For the studies that selected umbrella species from a set of candidates, we only recorded the final and optimal selections.

We recorded study locations, the year of publication, and the umbrella species recommended from each retained article. We analyzed the number of studies conducted on each continent that were published each year to reveal the spatial and temporal distribution of the study efforts and identified the representative umbrella species for each continent by counting the number of studies on each species. Then, each umbrella species’ threatened category at the time when it was first proposed was recorded by searching the IUCN (International Union for Conservation of Nature) Red List of Threatened Species, published in the year that was closest to the year of the species’ first recommendation as an umbrella species. We also recorded their threatened categories, which were assessed by the latest version of the IUCN Red List [24], and compared the potential changes from their first recommendations. We obtained the biological information and current conservation status of each species from the latest IUCN Red List, including taxonomy, population trend, habitat, threats, general use and trade, the conservation actions already in place, and the conservation actions needed. We analyzed the proportions of recommended umbrella species across different biological features and conservation statuses to reveal global trends in the selection of umbrella species.

In addition, we downloaded the distribution polygons of global vertebrates from the IUCN dataset [24]. To generate a global distribution map with higher certainty for each species, we trimmed each species’ ranges according to their attributes [25] and retained the sections whose presence was extant, had a native origin, or were reintroduced, introduced, or had assisted colonization; were seasonally resident or resident through the breeding season, non-breeding season, and passage. Then, each species’ distribution range was clipped to the Earth’s land surface using a layer of land range from the World Borders Dataset 0.3 (https://thematicmapping.org/ (accessed on 20 June 2022)) as a mask. The species that were extinct and extinct in the wild were removed as their spatial data were not available in the IUCN dataset. Finally, we compiled geographic ranges for 6685 amphibian, 6979 reptile, 10,909 avian, and 5647 mammal species. To determine if the recommended umbrella species had larger geographic ranges than global terrestrial vertebrate species, we compared the terrestrial ranges of recommended umbrella species with all vertebrate species using a two-tailed Mann–Whitney U test as the data were non-normal (Kolmogorov–Smirnov tests, both *p* < 0.001). We applied a rejection criterion of α = 0.05. Species distribution polygons were processed using ArcGIS 10.6 (ESRI, Redlands, CA, USA) under the World Mercator projected coordinate system, and statistical analyses were conducted using R 4.2.2 [26].

## 3. Results

### 3.1. Geographic Distribution

In total, we summarized 242 articles published since 1984 that were directly related to umbrella species. These articles collectively recommended 213 terrestrial vertebrate species as umbrella species (refer to Appendix A for a full list). The first scientific article with an umbrella species as a study species indexed by the Science Citation Index Expanded was published in 1995, after which the total number of studies and newly recommended umbrella species increased rapidly over time and peaked in 2019 (Figure 1A). The research was dominated by studies from North America, followed by Europe and Asia. There were fewer studies from Africa, South America, and Oceania and none from Antarctica (Figure 2). Consequently, there were considerably more umbrella species recommended in North America, Europe, and Asia than in other continents (Figure 2). The sage grouse (*Centrocercus urophasianus*) was the most extensively studied among the 213 species, as a representative umbrella in North America, followed by the tiger (*Panthera tigris*) in Asia and the western capercaillie (*Tetrao urogallus*) in Europe. The jaguar (*P. onca*) was a popular umbrella species in both North and South America.

### 3.2. Biological Features

Birds (46.9%) and mammals (45.1%) dominated the 213 recommended umbrella species, whereas reptiles (5.2%) and amphibians (2.8%) were rarely selected (Figure 1B). The distribution polygons were not available for four reptile species; thus, we calculated the terrestrial range area of 209 recommended umbrella species. The Persian fallow deer (*Dama mesopotamica*) had the smallest range area of 614 km^2^, whereas the migratory peregrine falcon (*Falco peregrinus*) and osprey (*Pandion haliaetus*) had the largest with over 90 million km^2^ ranges (including all seasonal ranges; Appendix A). The median terrestrial range for the recommended umbrella species (4,365,049 km^2^) was significantly larger than the median range of all terrestrial vertebrates (80,756 km^2^; Mann–Whitney U test, U = 1,140,467, Z = −15.942, *p* < 0.001). Over 70% of the 209 species had ranges above the upper quartile of all terrestrial vertebrate ranges (760,527 km^2^). In total, the 213 recommended umbrella species used 17 types of habitats (Appendix A), but a higher proportion of them occupied one to three types of habitats (Figure 1C), with forest, shrubland, and grassland being the most frequently used habitats (Figure 1D).

### 3.3. Threats and Conservation Statuses

The majority of the 213 recommended umbrella species faced different types of historical, on-going, and/or future threats (Figure 3A), with biological resource use, agriculture and aquaculture, and residential and commercial development being the most common threat types (Figure 3B). Generally, whole individuals, parts of individuals, or the products from individuals of those recommended umbrella species were consumptively used by humans for a diverse set of use and trade purposes (Figure 3C), with pets, food, and sport hunting as general types of end uses (Figure 3D). However, about 20% of species were neither threatened nor used, and most of them were least concern (LC) species (Appendix A).

The threatened categories for nine recommended umbrella species (4.2%) were unknown at the time they were the first recommended, but all of them were assessed in the latest version of the IUCN Red List (Figure 4). At the time of first recommendation, LC species accounted for about half of the 213 recommended umbrella species, followed by vulnerable (VU) species, while critically endangered (CR) species were the rarest (Figure 4). The threatened categories of most species remained unchanged from the time that they were first proposed until the latest version of the Red List (Figure 4), but eight species were downlisted, and five species were uplisted since their first recommendation (Appendix A). The global populations of most umbrella species were estimated to be currently decreasing (63.9%), whereas only 16.4% and 13.1% of species had stable or increasing populations, respectively. The population trend was not estimated for 6.6% of the umbrella species. 

Over half of the recommended umbrella species were protected under three or four types of conservation actions (Figure 5A), and land or water protection and management are currently in place for 205 species (Figure 5B). Based on specialists’ assessments, over 70% of these umbrella species will need more or improved conservation actions in the future (Figure 5C), and land or water management and protection are still required by a large proportion of species (Figure 5D).

## 4. Discussion

### 4.1. Geographic Bias in Umbrella-Focused Research

Since the 1984 inception of the umbrella-species concept [15], a considerable number of studies targeting umbrella species have been undertaken. We found that there was a rapid increase in published studies before 2019 (Figure 1A). This likely reflects the concerns of the global community for conserving biodiversity and the use of umbrella species as popular conservation strategies. This increase in publications may also reflect the availability of funding for these strategies. It is hoped that the increase in published studies has equated to an increase in on-the-ground conservation actions. Nevertheless, it is likely that the number of empirical umbrella-focused studies, on-the-ground efforts, and publication declines since 2019 was due to global constraints in field studies and conservation funding shortages resulting from the SARS-CoV pandemic [27,28], and the recovery of study efforts will probably lag behind the global post-pandemic recovery.

Our analysis has demonstrated a considerable geographic bias in study efforts toward the Northern Hemisphere. The number of studies was not in agreement with the global diversity of terrestrial vertebrates, with Asia, South America, and Africa indicating the highest species richness [29,30] but with lower study efforts. This is similar to other studies that have found a North American and European bias in the literature, probably correlated to the density of publishing conservation scientists [31,32]. Di Marco, Chapman, Althor, Kearney, Besancon, Butt, Maina, Possingham, Rogalla von Bieberstein, and Venter and Watson [31] also found that approximately 40% of national or sub-national conservation studies published between 2011 and 2015 were from the United States, United Kingdom, and Australia. It appears that conservationists in Oceania are publishing more non-umbrella-related articles. Therefore, the observed bias is likely due to the density of publishing scientists rather than any reflection of biodiversity-related factors. Perhaps the umbrella-species concept is less widely considered a viable option in these continents due to their focus on other methods of protection (e.g., ecosystems and habitats). However, it is likely that for several areas with rich biodiversity in the Southern Hemisphere, especially those developing countries in global biodiversity hotspots [33], conservation funding is limited, and thus, cost-effective strategies are key for greater conservation outcomes [28]. Given the proven usefulness of umbrella species for conservation [14,34], policymakers in these countries should take these strategies into account when formulating conservation practices and designing protected areas, and there are already diverse approaches that can provide informed guidance for selecting umbrella species with higher effectiveness and representativeness in a given region [35,36,37].

### 4.2. Taxonomic Bias in Recommending Umbrella Species

We found that amphibians and reptiles were infrequently considered or chosen as umbrella species compared to birds and mammals. This strong taxonomic bias in umbrella species was also reported in a previous review [6] and a meta-analysis [34]. This consistent preference for birds and mammals as umbrellas over amphibians and reptiles is likely due to several main factors. Firstly, compared to the majority of amphibians and reptiles, birds and mammals might meet some conventional criteria of umbrella species more easily, such as large body size, large home range, and ease of monitoring [38], making them more likely to be preconceived as capable umbrella species. Secondly, our deficient understanding of amphibian [39] and reptile distributions [40] has probably limited their usefulness as umbrella species and hindered any selection process. In addition, humans have had a long-standing preference for charismatic animals with aesthetic appeal [41,42]. Several of the most popular umbrella species also act as flagship species (e.g., the tiger and giant panda) and therefore need to be charismatically appealing. However, previous studies have suggested that satisfying conventional criteria does not necessarily lead to better umbrella effects, and a rigorous assessment of umbrella species is required to ensure their conservation effectiveness [34,38]. Given that umbrella species generally provide more benefit to same-taxon co-occurring species due to their similar resource requirements and shared threats [43], amphibian and reptile umbrellas could potentially provide better protection to sympatric amphibians and reptiles than currently favored avian and mammalian umbrellas. For example, water pollution and drought are generalized threats for amphibian species because they are so dependent on water quality compared to other terrestrial vertebrates, so the management of water sources in habitats targeting an amphibian umbrella species should confer direct conservation benefits to sympatric amphibian species [44]. Consequently, we recommend that more amphibians and reptiles should be assessed for their appropriateness as umbrella species across habitats and ecosystems.

### 4.3. Cautions in Selecting Umbrella Species

We found that a large proportion of existing recommended umbrella species are undergoing global population decreases and facing diverse external threats, and thus, most of these species are under protection and/or need further conservation. According to IUCN’s documentation [24], a few recommended umbrella species currently lack direct and formal (or widely recognized and recorded) conservation actions (Appendix A). Moreover, only a few umbrella species have been downlisted since their first recommendations due to successful conservation, and some have even been uplisted. Such disparities between scientific research and actual conservation practices using umbrella-species strategies are concerning, as an umbrella species can only open its conservation umbrella to shelter co-occurring species if it has been protected. 

A good understanding of natural history and ecological information are predictors of how well an umbrella species can represent and confer conservation benefits to ecosystems and co-occurring species [45]. However, we found that basic and important information, such as geographic distribution, population trends, and habitat, are not available for several existing umbrella species. Moreover, IUCN assessments for several recommended umbrella species have not been updated in a long time. The most recent IUCN assessments for 13 umbrella species were published ten years ago, and worse, the little-known and/or less-concerned species, such as the European pond turtle (*Emys orbicularis*) and gopher tortoise (*Gopherus polyphemus*), have not been evaluated since 1996 (Appendix A). This lack of up-to-date knowledge largely impedes the conservation practices targeting these species, such as the designation of protected areas, habitat protection, and population management, and could eventually reduce the usefulness and effectiveness of these little-known species as conservation umbrellas. Therefore, timely updates of IUCN assessments on those less-concerned species and taxa are warranted to fill knowledge gaps and address the lag in critical information.

We found that the 213 recommended umbrella species generally had large ranges. This is unsurprising, given that many umbrella species are chosen for their wide distributions, which encompass those of many co-occurring species [6,18]. However, wide-scale protection calls for vast resource investment [46] and international collaboration for conserving umbrellas throughout their extensive ranges; these are unrealistic requirements in most circumstances [43]. Alternatively, Ward, Rhodes, Watson, Lefevre, Atkinson, and Possingham [22] found that choosing a mix of traditional wide-ranging umbrella species and species with narrower ranges provided an increase in management efficiency within a set budget. In addition, given that the umbrella effect of a species may vary across spatial scales and sites, especially for migratory species that have distinct breeding and wintering ranges, a species with a satisfied umbrella effect in tested regions does not always perform well across its whole range [38]. Therefore, to maximize conservation outcomes while minimizing investment, a trade-off between wide- and narrow-range species should be taken into consideration when deciding which umbrellas are used for conservation schemes. Moreover, a rigorous assessment of potential umbrella effects is necessary prior to conservation practices in a new site.

Umbrella species appear to be most successful when they have similar threats with co-occurring beneficiaries, and thus, conservation actions targeting umbrellas could mitigate those shared threats [22,47]. However, we found that non-threatened species under the LC category dominated the recommended umbrellas of vertebrates. LC species may be more advantageous as umbrellas because they have relatively large populations that occur continuously (or nearly continuously) over their ranges [48], potentially enhancing active monitoring and management, especially when only one species is chosen as an umbrella [18]. However, species in this category are generally facing fewer threats. A large proportion of LC species that were recommended as umbrellas were neither threatened by any factor nor used by humans for any purpose. These species could probably not represent common threats of sympatric species very well, and their efficacy as umbrellas might be limited. Therefore, particular concerns about a species’ representativeness for the common threats to ecosystems or communities, besides their overlaps with regional biodiversity [49], should be raised when non-threatened species are selected and used as conservation umbrellas.

## 5. Conclusions

As an attainable strategy for maximizing conservation outcomes, the umbrella-species strategy has been studied for about 40 years, and 213 terrestrial vertebrate species have been recommended as umbrella species from 242 scientific articles published since the concept’s inception. We found a considerable geographic bias in global study efforts toward the Northern Hemisphere and also a remarkable taxonomic bias (toward birds and mammals) in the selection of umbrella species. In addition, wide-ranging and non-threatened species were preferred as umbrella species. Based on these revealed biases and trends, we have made some recommendations for the better application of umbrella-species strategies in the future. First, when conservation resources and funding are limited, policymakers in developing countries, especially those in the Southern Hemisphere, should not overlook umbrella species as viable strategies for maximizing conservation outcomes. Second, it is important to rigorously evaluate the effectiveness of different umbrella species with varying ranges and threats prior to conservation practices in a new site, and researchers should especially investigate the potential and appropriateness of amphibians and reptiles as umbrella species. Finally, the inadequate and/or outdated information and conservation statuses of umbrella species could impede their successful applications; therefore, IUCN specialists should update species assessments in a timely manner to fill our knowledge gaps and facilitate related conservation practices.

## Figures and Tables

**Figure 1 biology-12-00509-f001:**
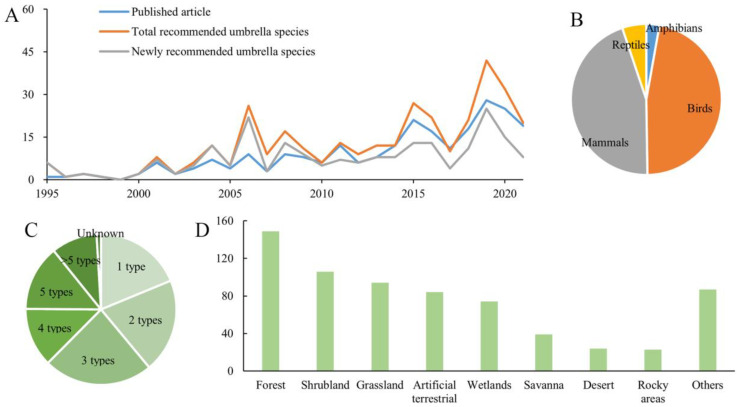
The yearly number of published articles with umbrella species as study species indexed by Science Citation Index Expanded, as well as the yearly number of total and newly (excluding umbrella species that had been proposed before) recommended umbrella species since 1984 to 2021 (**A**), the proportions of 213 recommended umbrella species across different taxa (**B**), the proportions of recommended umbrella species that use varying types of habitat (**C**), and the number of species for each type of habitat (**D**).

**Figure 2 biology-12-00509-f002:**
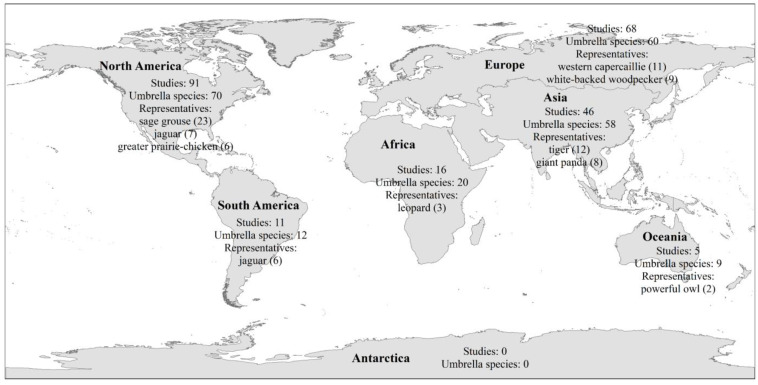
Distribution of study efforts and recommended umbrella species across continents, and the most studied representatives for each continent, with the number of articles on each representative in parentheses.

**Figure 3 biology-12-00509-f003:**
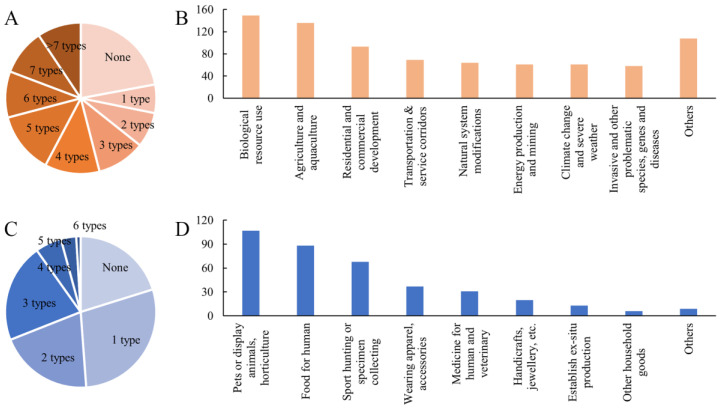
The threat statuses of 213 recommended umbrella species: (**A**) the proportions of recommended umbrella species that are impacted by varying types of threats and (**B**) the number of species that are impacted by each type of threat; (**C**) the proportions of recommended umbrella species that are harvested by humans (whole individuals, parts, or products from individuals) for varying types of end uses, and (**D**) the number of species that are used for each type of end use.

**Figure 4 biology-12-00509-f004:**
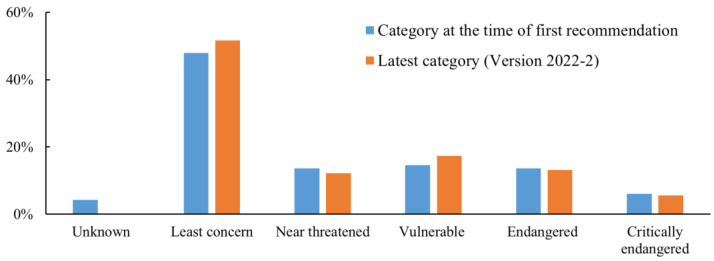
Proportions of the 213 recommended umbrella species across different threatened categories assessed at the year closest to the first recommendation of each umbrella species (blue bars), and assessed in the latest version of the IUCN Red List of threatened species (2022-2; orange bars). Any categories assigned as unknown, not recognized, and data deficient were grouped as unknown.

**Figure 5 biology-12-00509-f005:**
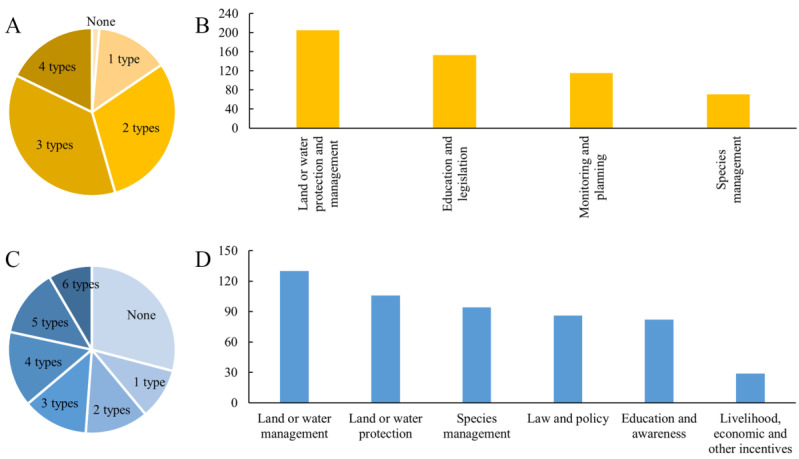
Conservation statuses of the 213 recommended umbrella species: (**A**) the proportions of recommended umbrella species that are currently protected by varying types of conservation actions, and (**B**) the number of species that are under protection for each type of conservation action; (**C**) the proportions of recommended umbrella species that need varying types of conservation actions, and (**D**) the number of species that need each type of conservation action.

## Data Availability

The data presented in this study are available in Appendix A.

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
