# Peer review of "Assessing Global Efforts in the Selection of Vertebrates as Umbrella Species for Conservation"

_biology, 2023, doi:10.3390/biology12040509_

Round 1

Reviewer 1 Report

This study summarizes global study efforts and recommended umbrella species for multi-species and community conservation. A total of 213 terrestrial vertebrates were identified as recommended umbrella species in 242 scientific publications since 1984, and their geographic patterns, biological features, and conservation status were analyzed to identify global trends in umbrella species selection. The findings revealed a considerable geographic bias of study effort from the Northern Hemisphere, a strong taxonomic bias in umbrella species selection, and a need to choose appropriate species for each location to ensure that the umbrella species strategy is applied appropriately.

My biggest concern with this manuscript is the methodology. The authors state that they systematically reviewed scientific articles. A systematic review is a rigorous and comprehensive approach to synthesizing existing research, which is not the case in this study. The search, screening, and selection process of included studies is NOT transparent nor repeatable.

The authors should use the PRISMA guidelines that help ensure the completeness and transparency of reporting in systematic reviews. The PRISMA diagram is an essential component of a systematic review because it provides a clear and concise summary of the study selection process. This can help readers assess the quality and completeness of the review.

In addition, the introduction and discussion sections are too brief. What is particularly missing is the comparison with other terms (e.g., keystone species, flagship species, etc.) in terms of their usefulness for species conservation.

Reviewer 2 Report

This is a very interesting review paper about geographic and biological trends in the study efforts and selection of conservation umbrella species. I found this ms very well written and appealing. There was a scientific needed for completing a kind of study like this one.

In spite of I have just a couple of minor comments that I hope can be useful to improve the paper.

Introduction

Figure 3. I suggest splitting this figure in two and improve the explanatory text of the figure.

Page 7, lines 219-220. I think authors are right, but I will try to avoid this kind of sentences “human preference for cute and furry/fluffy animals and our paucity of amphibian and reptile distribution data”. Therefore, I will delete this mention to the “furry animals”, and the mention of the lack of distribution data for amphibian and reptiles, that it is also repeated in page 7, lines 224-225.

Reviewer 3 Report

Please see uploaded document - I do not care to repeat myself.

Round 2

Reviewer 1 Report

I am satisfied with the revisions provided by the authors. Nevertheless, the Figures should be improved. It is generally not recommended to use pie charts in scientific publications and the meaning of the labels used (0, 1, 2, ...) is not clearly described.

Reviewer 3 Report

I really appreciate the authors' careful attention to reviewer comments and agree that the MS is much improved.

Besides the few specific comments / writing suggestions (on the pdf itself), here are some general comments for each section.

ABSTRACT

Do not change anything - it reads really well.

INTRO

I recommended a few minor changes, but I also do not see any major change to strengthen the ending. Originally, I had suggested "Second, the ending of the intro is quite weak – it is rather vague and, well, not impactful. Could the authors, for example, end with a brief description (and concrete example) of what they think the consequences of any biases or inadequate species selections might be." I do not see that this is done, but still think it could help the intro pack more of a punch. Maybe a statement starting with something like "taxonomic and/or geographic imbalances in the identities of the species that are proposed as umbrella taxa could exacerbate.... and thereby undermine effective conservation policy."

MATERIALS & METHODS

I completely understand and accept that the authors did not perform a systematic review and can accept their statement that a legit PRISMA statement may therefore be inappropriate (appreciate their explanation). That said, the reader would likely find it much easier to follow the 1st paragraph of the section (and remember what the authors did) with a PRISMA flow diagram. The website is back up, and PRISMA flow diagrams can be and have been used to provide this type of visual clarity for studies that are neither systematic reviews nor meta-analyses. For example, I made one for my recent, peer-reviewed, book chapter :-).

Otherwise, I find the methods section much better !

RESULTS

Wow. Much better. The authors asked me (in their response document) about the caption for fig. 1, specifically what I suggest for habitat types. They say "C. The number of habitat types means how many types of habitat a species uses. E.g., if a species uses forest, shrubland and grassland, its number of habitat types is 3; any forest obligate species’ number of habitat types is 1. Yes, we feel it’s hard to state clearly. Do you have some better suggestions".

Actually, I think the part I just made red above and highlighted is a pretty clear and perfect way to say it - just use that.

DISCUSSION

It reads really well now (just suggest a few minor corrections)

CONCLUSIONS

Good ! However, there were a couple of things I either questioned or did not understand. One is why only the Southern Hemisphere (see comment on pdf). The other is what the authors mean by "varying features" - this could really mean anything. My only other suggestion is that instead of having their three recommendations separated by semi-colons in a SUPER long sentence, why not have three separate sentences ? As in... First, when conservation resources and funding... Second, it is important to... Finally, inadequate and/or outdated... 

Incidentally, as an IUCN SSC member - I truly appreciate what the authors say. They are correct - RL assessments for many taxa are badly outdated and there are way too many DD (data deficient) taxa - believe me - I am a bat specialist - the DD problem is very severe for bats. That said, I hope the authors realise that RL assessors are really all volunteers. Not only is it super hard to do all these updates (and, to be honest, the IUCN did not even meet its own objective as far as how many assessments were supposed to be completed (never mind updated) by 2020), but also, for updates to be made, it takes good, on-the-ground or modeled data - and these are often lacking as well. I am not making excuses (again, I am not an Assessor), I am just saying that even though the authors are correct, and even though the will is there throughout the IUCN, it is a very, very steep uphill climb. I do not know what the authors' own taxonomic specialisations are, if any, but I would encourage them to get involved as assessors - the more bodies we have doing this work, the easier it becomes to accomplish these goals. 
